

# Soil carbon, available nutrients, and iron and aluminium crystallinity vary between boreal closed-canopy forests and open lichen woodlands

Carole Bastianelli[1,2,3,4], Adam A. Ali[4,5,6], Julien Beguin[2], Yves Bergeron[4], Pierre Grondin[7], Christelle Hély[3,4,6], and David Paré[2]

1 AgroParisTech, 16 rue Claude Bernard, 75005 Paris, France
2 Natural Resources Canada, Canadian Forest Service, Laurentian Forestry Centre, 1055 du P.E.P.S., P.O. Box 10380, Stn. Sainte-Foy, Québec, QC, Canada G1V 4C7
3 EPHE, PSL Research University, 4-14 rue Ferrus, 75014 Paris, France
4 NSERC/UQAT/UQAM Industrial Chair in Sustainable Forest Management, Université du Québec en Abitibi-Témiscamingue, 445 boul. de l'Université, Rouyn-Noranda, QC, Canada J9X 5E4
5 Université de Montpellier, Place Eugène Bataillon, 34095 Montpellier, France.
6 Institut des Sciences de l'Évolution de Montpellier (ISEM), Université de Montpellier, Place Eugène Bataillon, 34095 Montpellier, France
7 Ministère des Forêts, de la Faune et des Parcs, Direction de la recherche forestière, 5700 4e Av O, Québec, QC, Canada G1H 6R1

*Correspondence to:* Carole Bastianelli (carole.bastianelli@agroparistech.fr)

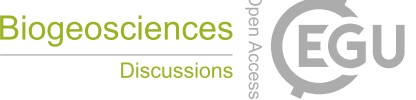

**Abstract.** At the northernmost extent of the managed forest in Quebec, the boreal forest is currently undergoing an ecological transition between two forest ecosystems. Open lichen woodlands (LW) are spreading southward at the expense of more productive closed-canopy black spruce-moss forests (MF). The objective of this study was to investigate whether soil properties could distinguish MF from LW in the transition zone where both ecosystem types coexist. All the soils studied were typical podzolic soil profiles evolved from glacial till deposits that shared a similar texture of the C layer. However, soil humus and the B layer varied in thickness and chemistry between the two forest ecosystems at the pedon scale. Multivariate analyses of variance were used to evaluate how soil properties could help distinguish the two types at the site scale. MF humus (FH horizons) showed significantly higher concentrations of organic carbon and of the main exchangeable base cations (Ca, Mg) than LW soils, which were nutritionally poorer. The B horizon of LW sites held higher concentrations of total Al and Fe oxides, and particularly greater concentrations of inorganic amorphous Fe oxides than MF mineral soils, while showing a thinner B layer. Overall, our results show that MF store three times more organic carbon in their soils (B+FH horizons, roots apart) than LW. We suggest that variations in soil properties between MF and LW are linked to a cascade of events involving the impacts of natural disturbances such as wildfires on forest regeneration that determines the of vegetation structure (stand density) and composition (ground cover type) and their subsequent consequences on soil environmental parameters (moisture, radiation rate, redox conditions, etc.). Our data underline significant differences in soil biogeochemistry under different forest ecosystems and reveal the importance of interactions in the soil–vegetation–climate system for the determination of soil composition.

*Keywords:* Boreal forest, soil properties, geochemistry, podzol, metal oxides, soil–vegetation interactions



## 1 Introduction

Vegetation–soil interactions are complex and constitutive processes of ecosystem dynamics, materialised by functional feedback roles between plant communities and the soil system (Richter and Yaalon, 2012; Van der Putten et al., 2013). Land-use changes or ecosystem shifts can have a wide range of impacts on soil properties such as nutrient availability, organic matter content, soil structure, erosion or soil water repellency (Li and Richter, 2012; Van der Putten et al., 2013; Willis et al., 1997). Concurrently, soil properties are of great importance for vegetation establishment and maintenance in space and time (Kardol et al., 2006).

In the central portion of Quebec's boreal zone, Canada, a southward expansion of open black spruce–lichen woodlands (hereafter LW) is currently being observed at the expense of more closed black spruce–moss forests (hereafter MF) (Bernier et al., 2011; Girard et al., 2008, 2009; Rapanoela et al., 2016). The current ecosystem shift could be due to a change in the regional fire regime (Ali et al., 2012; Rapanoela et al., 2016) that likely occurred several thousand years ago (Richard 1979; Asselin and Payette, 2005) and constitutes a hot stake raised by forest ecologists and forest managers in so far as open black spruce–lichen woodlands are less productive and, consequently, sequester less carbon than closed moss forests (Rapanoela et al., 2016; Van Bogaert et al., 2015). MF are characterised by dense stands mainly composed of black spruce (*Picea mariana*) with a ground layer dominated by feathermosses *(Pleurozium schreberi* and others) and sphagnum *(Sphagnum* spp.*)*. In LW, black spruce also stands as the dominant and quasi-exclusive tree species, yet tree density cover is a lot scarcer and the predominant ground cover vegetation is composed of lichens, mostly *Cladonia* spp.

The main factors influencing soil formation, namely climate, organisms, topography, parent material and time, also determine soil properties (Jenny, 1994; Lundström et al., 2000; Mourier et al., 2010). Soil formation is polygenetic and sensitive to ecosystem and vegetation changes (Richter and Yaalon, 2012). Transformation can be perceptible within a few years or decades, as it has already been observed following land-use changes (e.g., Li and Richter, 2012; Richter and Yaalon, 2012). Some vegetation communities have commonly been observed in association with specific soil types (Lundström et al., 2000, Mourier et al., 2010, Willis et al., 1997). Podzols are typically found in boreal spruce forests under well-drained conditions (Sanborn et al., 2011; Ugolini et al., 1981). Yet it is still unclear (i) what part pre-existing soil properties play in the establishment and maintenance of the different ecosystems, and (ii) how the persistence of a specific vegetation cover contributes to changing or maintaining specific soil properties. It is likely that both mechanisms coexist by relying on many feedbacks.

We have several reasons to hypothesize that ecosystems with different ground vegetation types, humus thicknesses and stand densities such as MF and LW could induce different soil geochemistry and soil horizon development. Differences in canopy openness and groundcover type may lead to variations in soil formation processes. MF soils should develop thicker accumulation (B) soil horizons as more organic matter inputs are provided by the denser vegetation, leading to higher nutrient availability (Bonan and Shugart, 1989; Haughian and Burton, 2015). Conversely, the low density canopy in LW should provide a lower nutrient supply, along with higher light and heat (Haughian and Burton, 2015; Sulyma and Coxson, 2001), leading to





different soil microclimatic conditions. Variations in hydrological processes are to be expected beneath both ground covers as moss layers, which have a high water holding capacity, create a saturated environment that is less favourable to decomposition, whereas lichen layers may maintain soil moisture at lower levels (Bonan and Shugart, 1989). Water fluxes, insulation properties and soil acidity may have a great influence on chemical reactions during pedogenesis, particularly redox conditions,

chemical element associations, and exchangeable cation circulation and mobilisation (Schwertmann, 1985; Brimhall and Dietrich, 1987). Lichens are also reported to be strong physical and chemical weathering agents of rock surfaces (Chen et al., 2000; Porada et al., 2014), while moss layer thickness may lead to a diffuse and weaker weathering. Differences in pedoclimate (Duchaufour, 1990) hydrological processes such as water fluxes, snowmelt rate, and drainage conditions (Buurman and Jongmans, 2005; Schaetzl and Isard, 1996), organic matter dynamics (Buurman and Jongmans, 2005) or weathering rate

(Lundström et al., 2000) could deeply impact the podzolization process.

Our main research objective was therefore to determine how soil biogeochemical properties and soil horizon development differ between two forest ecosystems, which are defined by different stand densities and ground vegetation compositions, but which both developed from glacial deposit in the transition region where both open- and closed-canopy forests occur as adjacent patches at the landscape scale.

We hypothesized that soils would be richer under dense forests with a moss cover that has higher C and N contents and higher base cation concentrations than lichen cover. Because iron complexation with organic compounds is a dominant process in podzolic B mineral horizons and is sensitive to properties such as soil pH, organic matter content or soil erosion (Holmgren, 1967; Li and Richter, 2012; McBride, 1987), we also hypothesized that iron reactive chemical species would be different depending on the local vegetation composition and aboveground density. We assumed that the conditions found in

soils covered by a lichen mat and low-density canopy would be more prone to iron and aluminium oxide accumulations than MF soils.

We therefore used an exploratory approach focusing on soil horizon thickness, carbon and nitrogen contents, base cation concentrations (Ca, Mg, K, Na), and species of iron (Fe) and aluminium (Al) (organically-bound and oxides) as elements of interest, and investigated their relations with vegetation properties. Our study was first conducted at the pedon scale for a

local approach (lichen vs. moss cover) on a 1-m² basis, and analyses were secondly run at the site scale (LW vs. MF) to assess whether local observations could be generalised to the ecosystem.

## 2 Materials and Methods

### 2.1 Study sites

The study area was located in the Canadian northern region of the Manicouagan crater in Quebec in the north portion of the

moss forest ecological domain (Fig. S1). Our study area covered approximately 900 km². Climatic data from the nearest reported station in Wabush Lake (52°55' N, 66°52' W, 551 m elevation, located 150-200 km northwest of our study sites) show a mean annual temperature of -3.1 ± 3.3°C and mean annual precipitation of 839.5 mm, with 51% falling as snow, for the



1981-2010 period. The annual sum of growing degree-days (GDD>5°C) was 1186.4 (Environment Canada, 2013). Black spruce (*Picea mariana* (Mill.) B.S.P.) was the dominant tree species in the area, but balsam fir (*Abies balsamea* (L.) Mill) was also scarcely found in MF. Groundcover vegetation was mostly composed of feathermosses (*Pleurozium schreberi* and others) with occasional patches of *Sphagnum* spp. in MF, while it was mostly composed of *Cladonia* spp. in LW. While vegetation in

the area was essentially composed of black spruce lichen−woodlands (LW) north of the 52nd parallel and of black spruce− moss forests (MF) in the south, patches of both ecosystems could be found in a patchy distribution throughout the area. The study area was particularly relevant for our comparative study in so far as it presented a patchy distribution of plots covered by moss and lichen developed from same deposit type (glacial deposit), within regional sites of LW and MF ecosystems. Types of deposit were yet slightly different from site to site (e.g. undifferentiated till, dead-ice moraine). We selected six independent

sites (experimental units or EU): three sites dominated by a MF ecosystem and three other sites dominated by a LW ecosystem (Table 1). Sites were preselected based on satellite images, and maps of the vegetation ecological domains and of the surface deposit types to focus on region 6P defined by the Ministère des Forêts, de la Faune et des Parcs du Québec (MFFP). Site selection was then validated through a field prospection. Each site was centred on a small headwater lake. Three transects were delineated per site; they extended from the lake shore out of the riparian zone up to 30 meters. Sample plots were placed along

each transect at distances of 10, 20 and 30 m from the end of the riparian area surrounding the lake, identified by the presence of *Chamaedaphne calyculata*. At each plot, soil samples were collected as described below. Stand characteristics were evaluated by measuring the basal area using a wedge prism relascope (factor 2) and by listing the measured tree species. Groundcover vegetation composition and abundance (in %) was estimated on a 1-m² area per plot prior to soil sampling; visual estimates were summed to 100%. Our design for a given forest type was therefore composed of six EU and nine sampling

plots per EU, for a total of 27 soil sampling plots per ecosystem type (MF vs. LW, 54 altogether). This experimental design was conceived for further investigations linking soil and lake sediment composition at the watershed scale. Indeed, it corresponds to the first step of a paleoecological investigation aiming to retrace the opening of the landscape over time using geochemistry analysis from lacustrine deposits. Before proceeding further, it was fundamental to test and demonstrate that nowadays the two types of ecosystem (LW and MF) display significant differences in soil properties at the watershed scale.

Although the ecosystem type was determined at the lake watershed scale, plots within a site could display non-typical cover vegetation. For instance, local lichen patches could be found in some plots belonging to the MF ecosystem, and vice versa. For practical purposes, lower case letters were therefore used when referring to vegetation type at the plot scale (lw and mf), while upper case letters were used when referring to the lake watershed/ecosystem (LW, MF). The most distant sites were 100 km apart. A general description of each site is given in Table 1. For statistical analyses including lichen and moss covers,

binary values were attributed to plots depending on the dominant vegetation cover type.

## 2.2 Soil sampling and treatment

Soils in the study area were typical podzols whose development in northern Quebec dated back to the colonisation of bare grounds by the boreal forest after glaciation 9,000 to 13,000 years ago (Lundström et al., 2000). Podzolization is favoured by



conifer stands and is formed by precipitation of litter organic matter that engages the mobilisation of soil Al and Fe (Buurman and Jongmans, 2005). A 20 x 20 cm$^2$ template was used to sample the organic layer (FH horizon) after discarding the live green moss/lichen portion. Mineral soil fractions were sampled in each horizon. They were composed of a thin eluvial pale grey horizon (Ae), a mineral B horizon whose colour varied between sites from light, reddish to dark brown and faded to paler

colours with depth, and a grey mineral C horizon. Soil samples of the mineral B and C horizons were collected in every plot (total N=54 for both forest types) in June and September 2015 and in one out of the three plots per transect (10 m from the lakeshore) for the FH horizon (N=18). In addition, the top 15 cm of the B horizon was collected volumetrically using a 5-cm diameter metal corer. Samples of the C horizon were retrieved with a soil auger as soon as the B-to-C horizon limit was reached. All soil samples were air dried, sieved at 2 mm, and weighed. This fraction was used for conventional analysis (pH,

texture) and for estimating bulk density. Samples were ground at 500 µm for geochemical analyses. C and N stocks as well as those of Al and Fe species were estimated and reported for the FH layer and the B top layer.

## 2.3 Geochemical analyses

### 2.3.1 Soil primary characteristics

Total C and N contents were measured on all soil samples by combustion using an induction furnace (Leco® CNS-2000

Analyzer). pH values and effervescence to HCl indicated the absence of carbonate. pH was determined by potentiometric titrations in deionized water. Soil texture was determined by soil fractionation and grain size sedimentation following instructions from Carter (1993).

### 2.3.2 Exchangeable ions and extractable phosphorous

Soil samples were treated with an extracting Mehlich-3 solution (CH$_3$COOH 0.2M, NH$_4$NO$_3$ 0.25M, NH$_4$F 0.015M, HNO$_3$

0.013M, EDTA 0.001M) at a 1:10 ratio (Mehlich, 1984). Concentrations of exchangeable ions and extractable phosphorous were then analysed by inductively coupled plasma atomic emission spectroscopy (ICP-AES). Cationic exchange capacity (CEC) was calculated as the sum of exchangeable cation concentrations (K, Ca, Mg, Mn, Al, Fe, Na).

### 2.3.4 Reactive Fe and Al chemical species

Podzols are characterised by a B horizon predominantly made up of amorphous material constituted of organic matter bounded

at different degrees to Al and Fe oxides and hydroxides (Canadian Soil Survey Committee, 1978; Lundström et al., 2000). Chemically-bound species of Fe and Al were extracted from mineral soil samples by means of three chemical methods, that rely on different reduction reagents: sodium pyrophosphate (Na$_4$P$_2$O$_7$), ammonium oxalate (C$_2$H$_8$N$_2$O$_4$) and dithionite-citrate bicarbonate (Na$_2$S$_2$O$_4$, Na$_3$C$_6$H$_5$O$_7$, NaHCO$_3$) (Pagé and Kimpe, 1989). We followed the extraction protocols of Mehra and Jackson (1960) and McKeague (1978). Concentrations of extracted ions were then analysed by ICP-AES. Species extracted

by pyrophosphate, oxalate and dithionite-citrate reagents will be respectively referred to as "pyro", "oxa" and "dit" hereafter.



Pyrophosphate is the weakest extractor, known to specifically isolate organically-bound iron ($Fe_{pyro}$). Oxalate removes $Fe_{pyro}$ as well as the inorganic amorphous iron ($Fe_{oxa}$). Dithionite citrate removes $Fe_{pyro}$, $Fe_{oxa}$ and crystalline iron ($Fe_{dit}$), i.e. most Fe species (Mehra and Jackson, 1960; Blume and Schwertmann, 1969; McKeague et al., 1971). Al species extracted by all three methods are reported to be of similar nature as Fe species, although extractions may be less specific and result in some overlaps (McKeague et al., 1971). In particular, quantities of Al extracted by oxalate ($Al_{oxa}$) may be higher than quantities extracted by dithionite citrate ($Al_{dit}$) in some cases such as in acid soils or podzols (Johnson and Todd, 1983; Pagé and Kimpe, 1989). Relying on iron extraction specificity, relative quantities of crystalline iron (e.g., goethite, hematite) can be obtained by subtracting the quantities of iron extracted with oxalate from those extracted with dithionite citrate ($Fe_{CRI} = Fe_{dit} - Fe_{oxa}$). Amorphous inorganic iron, also designed as short-range order mineral phases, is calculated by $Fe_{SRO} = Fe_{oxa} - Fe_{pyro}$ (Johnson and Todd, 1983; Pagé and Kimpe, 1989). Finally, we computed the commonly used iron crystallinity index ($CI = Fe_{dit} : Fe_{oxa}$) in order to asses soil development rate (Arduino et al., 1984; Blume and Schwertmann, 1969). Amorphous iron species gradually aggregate into crystalline forms during the pedogenesis process, which makes them good indicators of soil age (Arduino et al., 1984; Johnson and Todd, 1983).

### 2.4 Statistical analyses

To test the influence of spatial scale on associations of soil properties and belowground geochemistry between forest types, we performed statistical analyses at two different spatial scales: one at the soil pedon scale (plot scale) and another at the lacustrine watershed scale (site scale). At the plot scale, we tested for differences in each soil property between lw and mf using a mixed analysis of variance (ANOVA) with the variables "Site" as random intercept, "Transect" nested into "Site" as random intercept, and the dominance of moss vs. lichen as fixed effect. All mixed models were fitted using the nlme R package (Pinheiro et al., 2016) in the R environment (R Core Team, 2013). We then assessed covariances among aboveground attributes of vegetation (stand basal area and percent cover of moss and lichen) and chemical variables using partial least squares canonical analysis (PLSCA) and cross-correlation tests. PLSCA identifies latent variables ("orthogonal canonical component") that maximise correlations between two sets of variables given symmetric roles. PLSCA also makes it possible to calculate the proportion of variance explained by each orthogonal canonical component of one set into the other and to determine the "intragroup community index" (Tenenhaus, 1998) that represents the weight of one variable in a set in explaining variations of variables in the other set. To this effect, we grouped the vegetation variables estimated at the plot scale (e.g., basal area, moss and lichen cover) into one set of variables and the soil geochemical variables into another set. PLSCA was performed using the plsdepot R package (Sanchez, 2012).

Then, to determine whether association patterns between vegetation attributes and geochemical soil properties at the soil profile level could be scaled up at the site scale, we performed a permutational multivariate analysis of variance (perMANOVA) to test whether geochemical soil properties differed among ecosystem types (LW vs. MF). Following the parsimony principle, we restricted the number of soil chemical variables in our analyses to the five elements ($Al_{oxa}$, $Fe_{SRO}$, $Al_{SRO}$, Ca and Fe) that had the highest intragroup community indexes relative to vegetation components in the PLSCA. Justification for this choice is





that variables that have low intragroup community index are loosely linked to variables of the other set (Tenenhaus, 1998). PerMANOVA performs a permutational MANOVA procedure (Anderson, 2001; McArdle and Anderson, 2001) on similarity matrices (here obtained using Euclidian distances) taking the hierarchical structure of the experimental design into account in order to test whether the locations of centroids differ among the groups of interest in multivariate space. The three tested factors were: 1) ecosystem type (LW vs MF) as fixed effect; 2) site as random effect (N = 6); and 3) transect as random effect nested within site (N = 18 or 3/site). Most importantly, we avoided any pseudo-replication issue by testing the effect of ecosystem type (LW vs MF) with the random variable "site" as the error term (ddl of error term = 4; see Table 5). Moreover, to ensure valid results, we normalized our data prior to analysis and used the Monte Carlo method to maximise the number of permutation combinations when estimating the p-value (Anderson, 2001; McArdle and Anderson, 2001). PerMANOVA was performed using Primer 6 software (Anderson, 2001; McArdle and Anderson, 2001). We also tested whether multivariate dispersions around each group's centroid was homogeneous between the LW and MF ecosystem types using the vegan R package (Oksanen et al., 2016).

## 3 Results and Discussion

### 3.1 Soil profile analysis

Soil profiles from the two vegetation cover types showed visually *prima facie* pedological differences at the plot scale. All horizons were thicker under the mf cover than under lw cover (Fig. 1a, Table 2). Colours, although not defined with a soil handbook, also differed among soil profiles: while we observed reddish to light yellow B horizon in lw plots, they were darker and browner in mf plots (Fig. 1b) and 1c). Colour hues of mineral horizons have been reported to reflect various species and concentrations of Fe oxides (Arduino et al., 1984; Schwertmann, 1985), known as indicators of soil age and soil weathering rate. Because all sites developed from a similar geomorphological base (glacial deposit), we thought that the differences observed could portend different developmental or formation processes depending on the vegetation cover. Regarding texture prospection, we found no significant differences (p-value>0.1) in the mean percentages of sand, clay and silt between mf and lw samples of the C horizon nor in the mean percentage of sand and clay of the B horizon (Table 1).

### 3.2 Soil chemical properties

Regarding the FH humus layer, C and N concentrations were 30% to 50% higher in mf plots than in lw plots (Fig. 2, Table 2). Concentrations of the two dominant exchangeable base cations (Ca, Mg) were respectively 2.5 and 1.5 times higher in mf humus than in lw humus (Fig. 2). This translated into direct consequences on humus cation exchange capacity (CEC), which was ~50% higher in mf than in lw. However, no difference in humus pH was observed between mf and lw (Table 2).

With regard to the B horizon, contrary to the FH horizon, we detected no difference in C or N concentrations between mf and lw plots (Fig. 3, Table 2). As previously reported in acid soils of North American boreal coniferous forests (Bonan, 1990), the N concentration in mineral soil was very low (0.5-0.8%). The B horizon appeared to have higher Ca and Mg



concentrations in mf than in lw, while other ions concentrations were similar between plot types (Fig. 3). Regarding the pH, mf B horizons were ~5% more acid than lw B horizons (Table 2).

Concerning the C horizon, as expected, C percentage was lower in this deep horizon than in the FH and B horizons (Table S1). Yet, the percentage of C was ~2 times higher in the C horizons of mf plots than in that of lw plots (Table S1). This organic enrichment can only have a biological origin coming from the upper layers as no C is provided by the mineral parent material in these acidic soils evolved from a granitic bedrock. Similarly, extractable phosphorous (P) was 2.5 times higher in the lw C horizon than in that of mf. The accumulation of products of mineral weathering as well as the migration of organic P compounds could explain this difference.

### 3.3 Fe and Al reactive species in mineral horizons

Our results for the different Fe and Al species in the B horizon show that ranges of organically-bound metal concentrations ($Fe_{pyro}$, $Al_{pyro}$) were similar in lw and mf plots (Fig. 4). However, lw B horizon exhibited 1.5 to 2.5 times higher concentrations of Fe and Al extracted by oxalate and dithionite citrate than mf B horizon. In compliance with other studies performed in acid forest soils (Johnson and Todd, 1983), we observed higher concentrations of $Al_{oxa}$ than $Al_{dit}$ (Fig. 4). The main difference between lw and mf B horizons lied in the proportion of inorganic amorphous Fe and Al ($Fe_{SRO}$, $Al_{SRO}$) (Fig. 4, Table 2) while crystalline iron concentrations ($Fe_{CRI}$) were similar (Table 2).

In the C horizon, no major variations in Al and Fe species concentrations could be observed between lw and mf plots (Table S1). This deeper horizon was also less concentrated than the B horizon in all reactive Fe and Al species, which is consistent with the findings that, in podzolic soils, metallic elements are more concentrated in the upper centimetres of the B horizon (Lundström et al., 2000). Because no differences in Al and Fe species concentrations were found in the C horizon as opposed to our observations in the B horizon, our result suggest that the B horizon's chemical structure and metal oxides composition may have been influenced by different pedogenetic development under mf and lw cover, rather than by the mineralogical origin of their parent materials. In particular, the absence of any noticeable difference in the iron crystallinity index (CI) of B horizon between all studied sites indicated that they had identical ages (Table S2). Structural and composition variations are thus derived from other drivers than soil origin and instead depend on factors influencing horizon formation processes.

Our results diverge from the observations made by Ugolini's et al. (1981) who found no morphological variations nor geochemical differences in the soil profiles of lichen tundra and spruce forest in a boreal zone of Alaska. In particular, they found no difference in reactive Fe species concentrations ($Fe_{pyro}$, $Fe_{dit}$) between the two ecosystems. They concluded that time and climate were stronger drivers of soil formation in their study area than the vegetation type. We suggest that because of the proximity of our study sites and the patchy plot distribution within sites, climate could be considered homogeneous in the present study. Our results are rather in line with those of Li and Richter (2012) who showed that land-use changes between old hardwood forests, cultivated agricultural fields and old-field pine forests can induce transformation and redistribution of soil iron oxides over relatively short timeframes ($Fe_{oxa}$, $Fe_{dit}$). They inferred that some differences could be due to the different





erosion levels and biological activities impacting soil iron oxides and organometallic compound transformation (Li and Richter, 2012).

### 3.4 Phosphorus distribution

We observed low concentrations of extractable phosphorous in the B horizon, contrary to the FH and C horizons (Figs. 2 and 3, Table S1). These results could be explained by P sorption properties to $Fe_{SRO}$ and $Al_{SRO}$ species. As a matter of fact, in the B horizon, most P is bound within Fe-P and Al-P complexes (Grand and Lavkulich, 2015; Li and Richter, 2012). Grand and Lavkulich (2015) showed that P sorption to short-range order Al and Fe mineral phases decreased the availability of labile P. In the C horizon, in addition to its enrichment in organic C, the smaller amounts of Fe and Al oxides may be the reason why more labile P is available (Table S1). Reactive Fe and Al species are known to play an important geochemical role in acidic forest soils due to their sorption properties that influence carbon and nutrients bioavailability through coupling reactions, which makes them good predictors of nutrient availability (Grand and Lavkulich, 2015; Li and Richter, 2012). Here, P sorption to short-range order Al and Fe mineral phases in the B horizon may reduce nutrient availability to plants, resulting in limited P supply. P limitation is commonly observed in acidic soils of boreal forests (Giesler et al., 2002).

### 3.5 Relations between B and C horizons

Soil properties were analysed by considering the local variability of the parent material and, therefore, by studying the ratio of B to C horizons for various properties. We found that B:C ratios were different between mf and lw soils for many chemical properties (e.g. Ca, $Al_{oxa}$, $Al_{dit}$, $Al_{SRO}$, $Fe_{dit}$, $Fe_{SRO}$; cf. Fig. 5), which suggests that dissimilar biogeochemical processes and vertical transfers occur locally in the soil of the two vegetation types. These differences in B:C ratios between mf and lw soils also confirm that C horizon composition is unlikely to drive most of the variation observed between lw and mf B horizons and that it is the influence of vegetation that impacts most of the soil biogeochemistry. Furthermore, the low concentration of chemical elements in the C horizon compared with B and FH horizons also invalidate the hypothesis of a deeper mineralogical influence explaining the main differences in geochemical composition between lw and mf plots. Finally, variations in soil conditions such as temperature, pH, soil hydrology, could play a role in differentiating horizon composition. The thickness of the organic layer and its higher water retention capacity in mf forests could greatly affect soil processes, and this is also reflected in the thickness of the B horizon.

### 3.6 Covariance between vegetation and soil geochemical variables

Results of the multivariate PLS canonical analysis conducted using the five most significant compounds (according to Figs. 3 and 4, Table 2) support the hypothesis of a different biological influence of vegetation on soil chemical composition and structure that discriminates between the two forest types.

Indeed, our results revealed that the five soil geochemical variables with the greatest intragroup community indexes were $Al_{oxa}$, $Fe_{SRO}$, $Al_{SRO}$, Ca et Fe (Fig. 6, Tables 3 and 4), meaning that they were highly linked to variables in the vegetation





set. Vegetation variables on the other hand all had an important weight in explaining the variability of soil physico-geochemical variables (Table 4). This result was consistent with stand basal area (representative of forest production) and cover type vegetation being tightly correlated (Fig. 6). Fe and Al complex species were positively correlated with each other, positively correlated with lichen cover, and negatively correlated with moss cover and basal area (Fig. 6). This result was consistent with

our aforementioned quantitative observations at the plot scale. Exchangeable Fe, Ca and Mg were positively correlated with each other, as well as with the vegetation characteristics of dense moss-covered stands (Fig. 6). Exchangeable Fe, Ca, Mg behaviours were very much alike, displaying negative correlations with Fe and Al oxides (Figs. 4, S2 and S3). The different behaviours of exchangeable Fe and bound Fe could be explained by their different mobility properties and abilities, in particular since fluxes could vary under different soil environmental conditions and thicknesses between lw and mf plots.

We also found positive correlations between C and N concentrations, moss coverage, and Mg and Ca concentrations in the FH organic horizon (Fig. S3). The higher base cation bioavailability in mf plots could be explained by greater inputs of organic matter to the soil surface and by a higher coniferous basal area cover. Decomposing organic matter and litter are known to be important sources of base cation supply such as Ca and Mg (Finzi et al., 1998; Grand and Lavkulich, 2015).

### 3.7 Differences in soil geochemistry at the site scale

We scaled up the effect of ground cover type (lw vs. mf) on $Al_{oxa}$, $Fe_{SRO}$, $Al_{SRO}$, Ca and Fe at the site scale (lake watershed scale) to test explicitly if the same chemical elements differed between ecosystem type (LW vs. MF) using perMANOVA. Our results showed significant differences between MF and LW ecosystem types (P(MC)=0.0005, pseudo-F=12.939). The type of ecosystem alone explained 47.6% of the total variance in the studied set of geochemical variables (Table 5). In addition, our test of multivariate dispersion revealed that the level of heterogeneity regarding $Al_{oxa}$, $Fe_{SRO}$, $Al_{SRO}$,

Ca and Fe did not differ between MF and LW (p-value=0.895, F=0.0203, Table S3), which reinforces the conclusion that the difference observed between MF and LW originates from differences in mean values of $Al_{oxa}$, $Fe_{SRO}$, $Al_{SRO}$, Ca and Fe rather than differences in variance (Figure 7). This finally highlights interactions between ecosystem structure and geochemical composition of the soil at the site scale.

### 3.8 Total element stocks in the soil

Because LW and MF displayed variations in soil composition and thickness, we hypothesised that they should hold different total amounts of chemical species. Total net stocks of C and N were scaled up from the plot to the site scale (layer thickness x organic matter concentration) in FH and B horizons (top 15 cm). On average, the FH horizon in MF held 3.5 times higher amounts of C and 4 times more N than that in LW (Table 6). Similarly, in the B horizon, both C and N stocks were 2 times higher in MF than in LW. These results suggest that, in addition to C sequestration in a greater aerial biomass, closed

moss forests also hold more C in their soil. However, regarding Fe and Al species, total net stocks were higher in LW than in MF, despite their thinner B horizons (Table S4).





### 3.9 Biological influence

Overall, because we did not found any difference in the geochemistry and texture of the C horizon between ecosystem types and because our sites developed from deposits of similar origin, our results, which are in line with those of other studies, suggest a biological influence of vegetation on soil profile development and soil chemistry. Haughian and Burton (2015) showed that more variability in soil composition could be explained by vegetation functional groups (e.g., mosses vs. lichens) than by abiotic characteristics (soil texture, topography). They found that variations in nutrient availability were the result of differences in vegetation types (lichens, feathermosses and vascular plants) rather than the opposite (i.e., nutrient availability as a cause of vegetation patterns). In their study of podzol biogeochemical vertical stratification in hardwood forests of New Hampshire, Wood et al. (1984) concluded that the B horizon is subject to strong geochemical control that is under biological influence rather than of mineralogical origin. Finzi et al. (1998) observed different distributions of Ca, Mg, Fe and Al exchangeable cations in 0-7.5 cm mineral soils under various tree species. They found an association between tree species and soil-specific chemical properties at the tree scale, and suggested that vegetation influenced soil acidity and cation cycling in the forests studied (Finzi et al., 1998). Here, we suggest that the differences we observed in soil geochemical structure and composition between MF and LW had a biological origin that resulted in repercussions of the local vegetation on soil environmental conditions, which in turn influenced soil biogeochemical processes. We suggest the following potential causes for the difference observed in soil between LW and MF ecosystems: i) the abundance of tree cover could influence soil formation through its direct consequence of canopy openness on soil micro-environmental conditions and drainage; ii) snowfall rate and snowmelt duration in turn may influence reductive conditions and cause differential dissolution of iron oxides that accumulate in the B horizon (Giesler et al., 2002); iii) thickness of organic soil layers could create differential insulating properties (Lawrence and Slater, 2007); iv) because water flow also affects the distribution of chemical elements such as labile P, Fe and Al (Giesler et al., 2002), differences in drainage conditions, moisture and hydrochemical processes under moss and lichen covers (Brown et al., 2010; Haughian and Burton, 2015; Price et al., 1997) could also explain the observed variations in lw and mf soil chemical properties; and finally, v) lichen surface weathering capacity (Chen et al., 2000; Porada et al., 2014) may be responsible for the higher concentrations of Fe and Al species in LW upper mineral horizons, while a dense moss cover may be less aggressive in mineral weathering; however because this environment is more productive, it could generate more organic acid and favour a deeper profile development.

### 3.10 Soil, climate and vegetation dynamics

It has previously been suggested that the two vegetation types considered could be two ecological states and that LW could be an alternative stable state resulting from regional disturbance history (Jasinski and Payette, 2005). We propose here a diagram (Fig. 8) that synthesises our interpretations of the possible feedback processes between climate, vegetation and soil biogeochemistry that can be considered to result from successive fires. Indeed, the progression of open canopy forests could be due to a greater fire frequency resulting from the changing climate (Rapanoela et al., 2016). Differences in fire events may



lead to direct and indirect consequences at the soil level arising from fire impacts on vegetation structure and soil properties (Certini, 2005). Fire effects on soil properties have been shown to range from negative short-term effects (removal of organic matter, erosion, loss of nutrients through volatilisation, alteration of microbial communities, etc.) to long-term consequences (enhanced productivity, impact on forest successions, etc.) (Certini, 2005). While negative effects on soil properties seem to
be short-lived (detectable some years post-fire, at most a decade) and restricted to a few top centimetres of superficial layers (Certini, 2005), we suggest that indirect effects may be wider and could have longer term consequences. Frequent fires may hinder the accumulation of a thick top layer of organic matter, thus leading to direct aftereffects on soil physical and microclimatic conditions. The low stem basal area (likely associated with lower tree density) inherited from frequent fires results in both higher radiative insolation and precipitation reaching the soil surface, thus setting an environment more suitable
to the establishment of light-tolerant lichen rather than colonisation by moss species. Repeated fires may spearhead lichen dominance in LW by maintaining preferential environment conditions for its colonisation and establishment (Girard et al., 2009). In return, lichen establishment could as well maintain a specific soil composition that is low in nutrients because of its low primary productivity (Moore, 1980). Altogether, the disturbance regime in boreal forests could determine the ground cover vegetation type and impact soil development through both direct and indirect effects, by generating poorer soils and sustaining
the establishment of less productive forests.

## 4 Conclusions

We identified clear relationships between soil and vegetation structures that are reflective of a whole integrative system relying on feedback interactions. Although the correlation patterns between the ecosystem's biological components and the soil variables seem complex, our results suggest that in comparison to closed canopy forests, open forests with a lichen
ground cover are associated with a soil impoverished in C and available nutrients that develops a thin accumulation B horizon characterised by high concentrations in amorphous species of Fe and Al. Ecosystem productivity and carbon sequestration are affected twice in LW compared with MF: through a lower density of trees (lower basal area and fewer stems) and through nutrient- and organically-limited soils. This emphasizes the current economic and climatic stakes that forest opening represents in so far as it could have important consequences in terms of carbon sequestration capacity. Our study of soil compartments
confirms that the current opening of black spruce forests is an ecological, economic and climatic stake with underlying long-term consequences, notably in terrestrial carbon budget.

A good understanding of the processes governing soil biogeochemistry and feedback interactions between soil and vegetation remains fundamental for forest management, especially in areas of ecological transitions representing major challenges, such as the northern boundary between productive boreal forests and open lichen woodlands. Our study highlights
that natural disturbances may influence landscape remodelling and ecosystem heterogeneity in much more ways than through their direct impacts on seedling regeneration and soil nutrient short-term depletion. Repeated disturbance events could have long-term consequences on soil formation and development. Disturbance history and inheritance could promote the

establishment and maintenance of specific vegetation-soil systems (or ecological states). If so, soil science and biogeochemistry could become interesting proxies in disturbance ecology, notably for paleoecological investigations aiming to reconstruct changes in vegetation. Further investigations should explore vegetation structure (basal area and tree density) and soil relations in other ecosystems and focus on microclimatic and drainage conditions as well as disturbance history as explanatory drivers of variability.

**5 Data availability**

The raw data compiled for this study are available at *[to be determined]*

**Author contribution**

AA, CB, DP, YB, PG and CH designed the research. AA, CB, PG and DP collected samples in the field. Experimental analyses were carried out by CB and Serge Rousseau. Results were statistically analysed by CB and JB. CB prepared the manuscript with input from AA, JB, DP, CH and YB.

**Acknowledgements**

        This research was funded by the Natural Sciences and Engineering Research Council of Canada (NSERC), the European IRSES NEWFOREST project, the Institut de l'Écologie et Environnement of the Centre national de la recherche scientifique (CNRS-InEE), the École Pratique des Hautes Études and the University of Montpellier (France) through the International Research Group on Cold Forests (GDRI "Forêts Froides", France), and the Institut Universitaire de France (IUF). The Ph.D. thesis of C. Bastianelli was supported by AgroParisTech.

        The authors are grateful to M. Serge Rousseau for carrying out laboratory experiments, to M. David Gervais for his help with field work and to Ms. Isabelle Lamarre for her help in proofreading this paper. *Les auteurs remercient également Véronique Poirier de la direction de la recherche forestière du Ministère des Forêts, de la Faune et des Parcs du Québec (MFFP) pour son aide dans l'obtention des données spatiales, ainsi que Pierre Clouâtre (MFFP) et Benoît Gaudreau (MFFP) pour leur apport logistique et implication dans la préparation du terrain.*





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


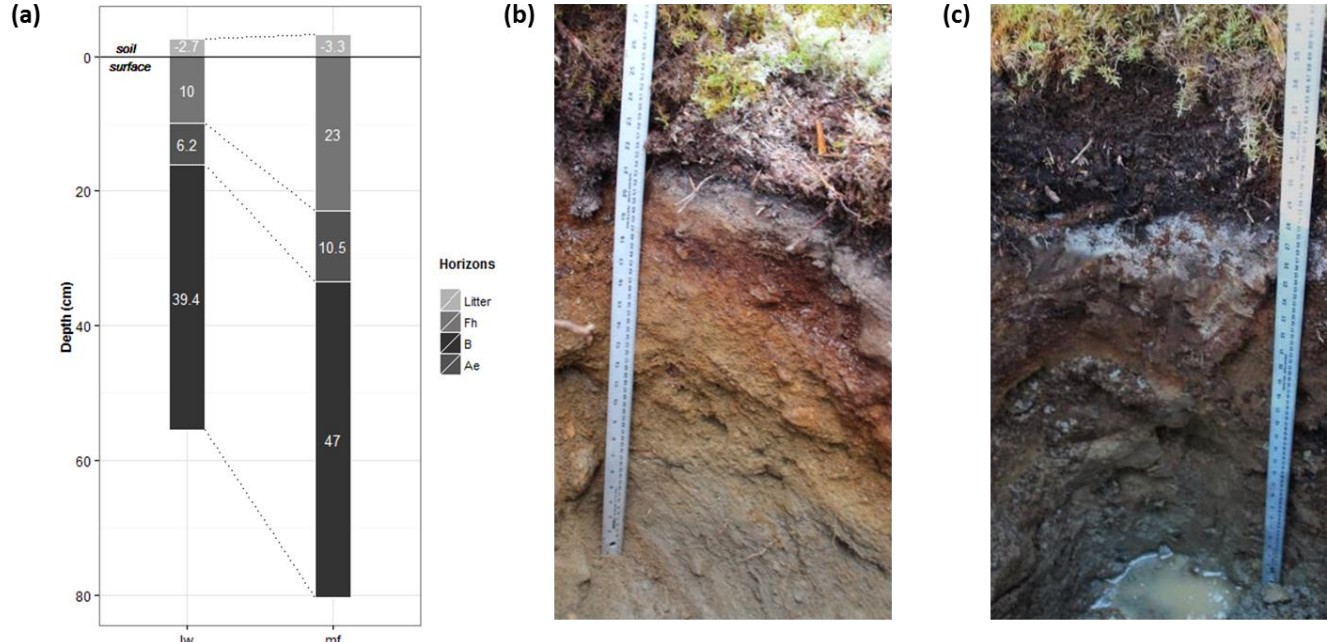

**Figure 1: Soil profiles in mf and lw plots.** (a) Average thicknesses of the litter, FH, Ae and B horizons. Labels inside stacked bars refer to each horizon's thickness average. (b) Picture taken in the field of a soil profile representative of lw. (c) Picture taken in the field of a soil profile representative of mf.



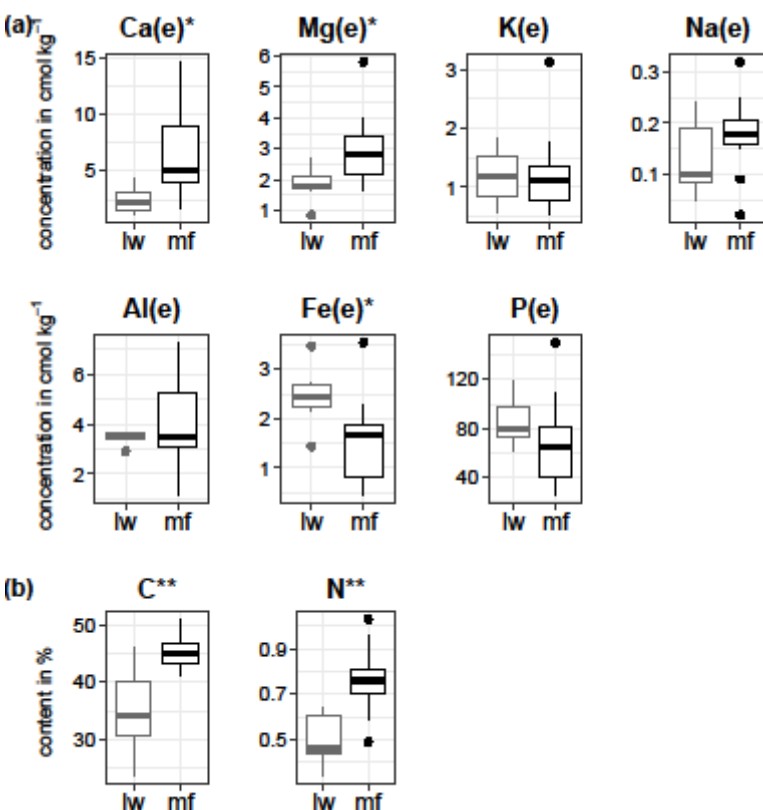

**Figure 2: Distributions of chemical compounds in the FH horizon in lichen woodland and moss forest plots (lw and mf, respectively).**
(a) Concentrations of exchangeable base cations and extractable P, (b) organic C and N. Boxplots represent the distribution around the median values. (e) indicates exchangeable or extractable elements. Asterisks indicate significant difference using mixed models (see Material and Methods), with *: p-value<0.05, **: p-value<0.01, ***: p-value<0.001.

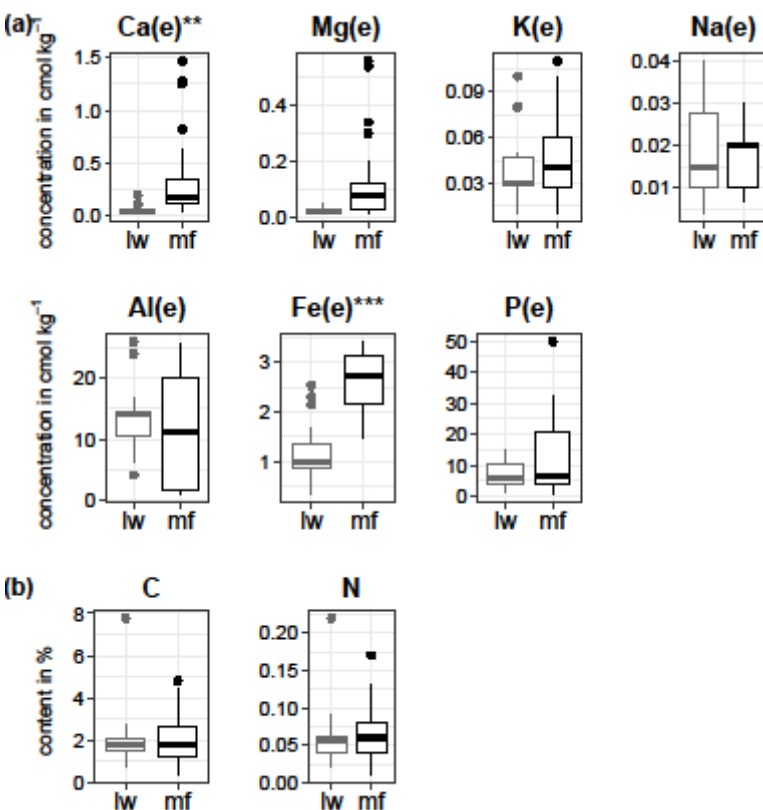

Figure 3: Distribution of chemical compounds in the B horizon in lichen woodland and moss forest plots (lw and mf, respectively). Same legend as in Fig. 2 applies for statistical significance.





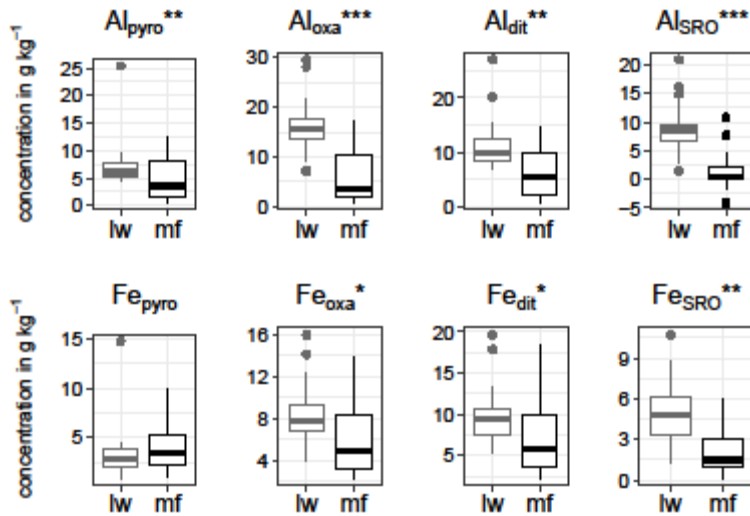

**Figure 4: Distribution of Fe and Al species concentrations in the B horizon of lw and mf plots.** Boxplots represent the distribution around the median values. Same legend as in Fig. 2 applies for statistical significance.





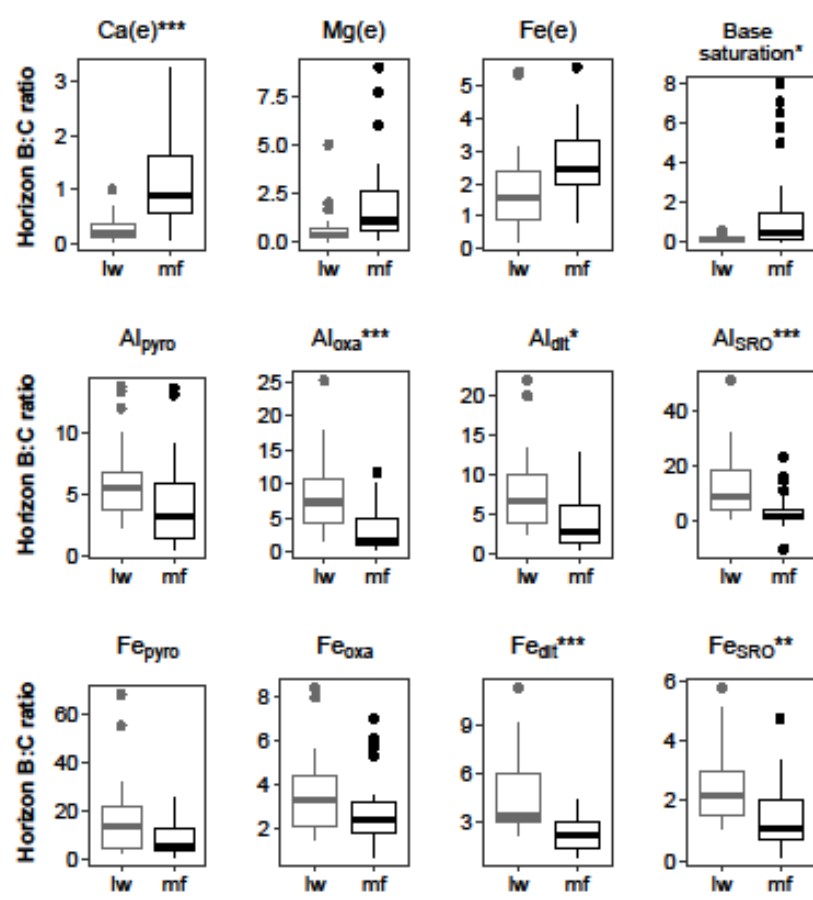

**Figure 5: Comparison of horizons B:C ratios in lw and mf plots.** Ratios were calculated by dividing for each soil type concentration measured in the B horizon by that measured in the C horizon. (e) indicates exchangeable or extractable elements. Asterisks indicate statistical significance using mixed models (see Material and Methods) with *: p-value<0.05, **: p-value<0.01, ***: p-value<0.001.




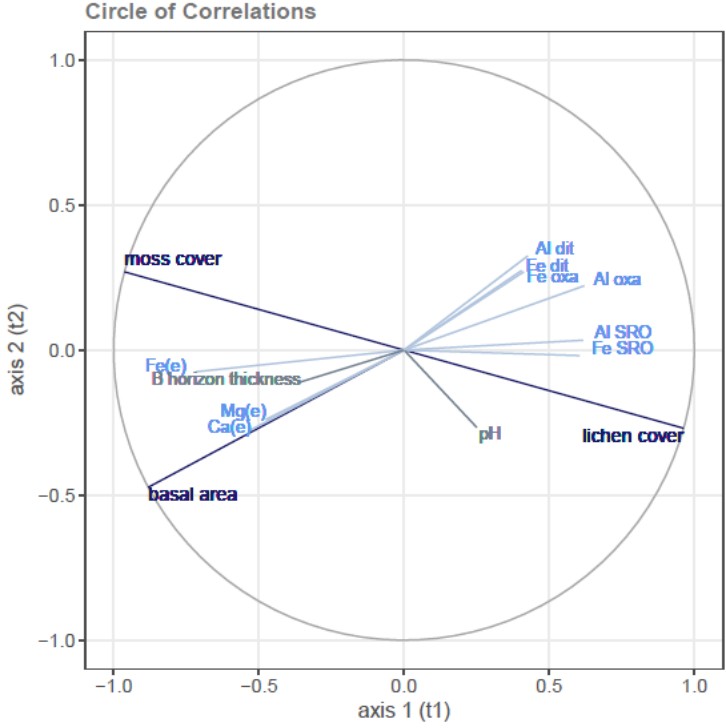

**Figure 6: Graphical projection of partial least square canonical analysis (PLSCA) results in the B horizon at the plot scale.** Variables for stand tree cover characteristics are drawn in dark blue, chemical compounds are in light blue, and soil characteristics in grey. Axes correspond to principal orthogonal canonical components. Positive, null or negative correlations between variables are indicated by acute, right or obtuse angles, respectively, between the corresponding vectors. (e) indicates exchangeable or extractable elements. Regarding the chemical elements, only variables that showed the greatest differences in mean values between lw and mf plots in the B horizon were included in the PLSCA.



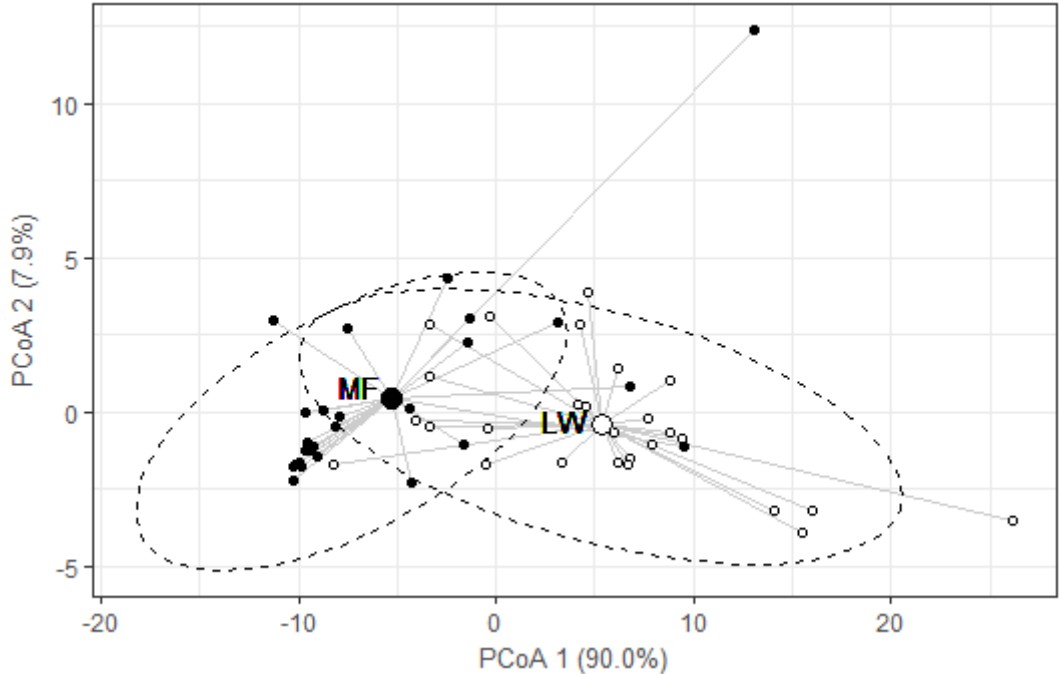

**Figure 7: Graphical projection of homogeneity of multivariate dispersions of data around ecosystem type centroids (Principal coordinates analysis).** Average distance to centroids were 6.084 and 5.912 for LW and MF, respectively. Small open circles represent plots located in LW sites, small black filled circles correspond to plots located in MF sites. The two big circles represent the group centroids for each forest type (LW vs. MF).




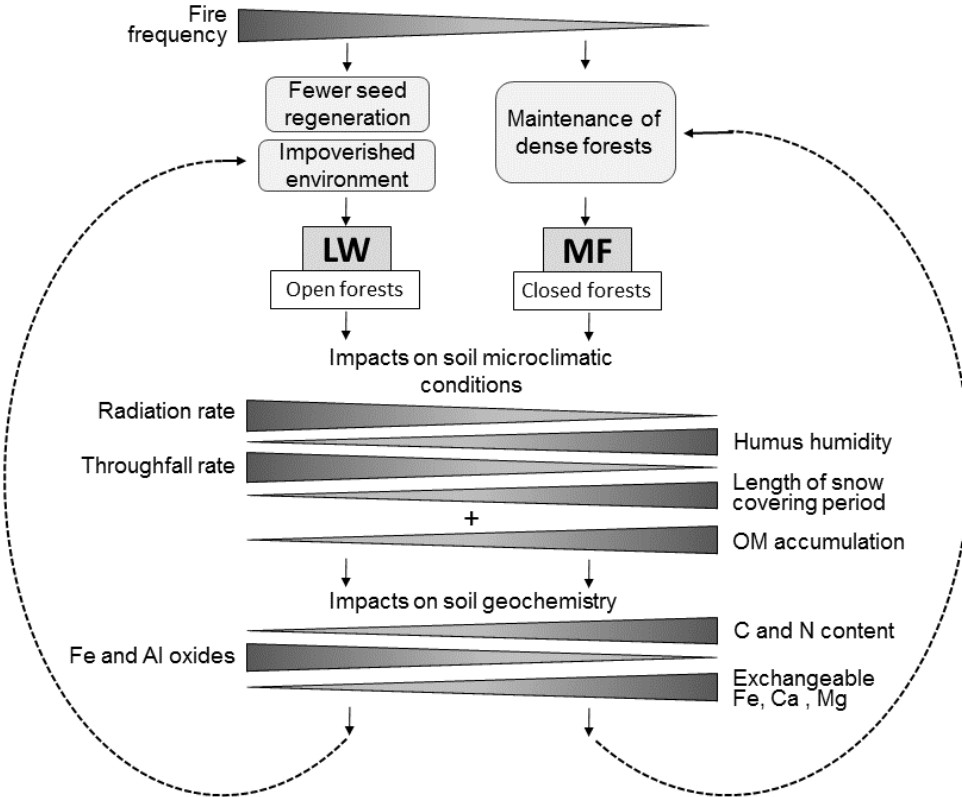

**Figure 8: Schematic illustration of feedback processes between stand biomass (basal area and/or density) and soil biogeochemistry as a consequence of climatic conditions based on the present study interpretations.**



**Table 1: General information and description of the six sites of study.** LW = lichen woodland, MF = moss forest. Values of basal area, ground cover composition and soil horizon thickness are given as means ± standard deviations to illustrate intrasite variability. Means where calculated from nine plots per site, regardless of the local ground cover type, but rather considering the regional ecosystem type of the site they belonged to. NA = not determined during field sampling because of a fire that burned the forest in 2007.

| Site | Coordinates | Eleva-Tion (m) | Site ecosystem type | Basal area (m².ha⁻¹) | Average soil horizon thickness FH (cm) | Average soil horizon thickness B (cm) | Average ground cover composition % moss | Average ground cover composition % lichen | Soil texture (%) (B horizon) sand | Soil texture (%) (B horizon) silt | Soil texture (%) (B horizon) clay |
|---|---|---|---|---|---|---|---|---|---|---|---|
| Lake Mundi | 52°15'6.293" N 67°42'44.681" W | 573 | LW | 8.2 ± 3.4 | 13 ± 6 | 36 ± 4 | NA | NA | 69.2±7.1 | 21.4±7.6 | 9.4±2.6 |
| Lake Prisca | 52°10'8.503" N 67°55'38.621" W | 563 | LW | 10.0 ± 4.1 | 13 ± 9 | 33 ± 7 | 18 ± 20 | 82 ± 20 | 68.9±5.7 | 22.8±5.8 | 8.3±1.1 |
| Lake Adele | 51°50'10.72" N 67°55'45.962" W | 457 | LW | 10.0 ± 4.3 | 14 ± 3 | 52 ± 8 | 3 ± 5 | 97 ± 5 | 72.6±13.9 | 22.1±13.1 | 5.3±2.1 |
| Lac des trotteurs | 52°53'24.9" N 68°12'6.999" W | 555 | MF | 25.8 ± 4.9 | 29 ± 12 | 48 ± 12 | 99 ± 2 | 1 ± 2 | 76.5±6.7 | 16.4±5.4 | 7.1±1.5 |
| Lake Arthur | 52°13'45.965" N 67°44'46.98" W | 661 | MF | 15.1 ± 4.6 | 19 ± 4 | 39 ± 15 | 83 ± 31 | 17 ± 31 | 76.1±6.6 | 16.8±5.9 | 7.1±1.3 |
| Lake Freeze | 51°49'44.513" N 68°0'45.545" W | 580 | MF | 18.9 ± 6.9 | 39 ± 13 | 57 ± 10 | 100 ± 0 | 0 ± 0 | 80.1±11 | 10.8±7.5 | 9±4.5 |

**Table 2: Characteristics of FH and B horizons in mf and lw plots.** lw plots = plots covered by lichen woodlands, mf plots = plots covered by moss forests. Values are given as means ± standard deviations. Asterisks indicate statistical significance using mixed models (see Material and Methods) with *: p-value<0.05, **: p-value<0.01, ***: p-value<0.001.

| | FH Horizon lw plots (n=7) | FH Horizon mf plots (n=11) | FH Horizon p-values | B Horizon lw plots (n=22) | B Horizon mf plots (n=32) | B Horizon p-values |
|---|---|---|---|---|---|---|
| Horizon thickness | 9.64 ± 4.44 | 22.25 ± 10.3 | 0.050* | 39.38 ± 12.77 | 47.07 ±12.07 | 0.048* |
| %C | 34.99 ± 8.07 | 45.41 ± 3.16 | 0.003** | 2.04 ± 1.36 | 2 ± 1.11 | 0.418 |
| C:N | 69.62 ± 5.9 | 61.97 ± 12.38 | 0.158 | 34.9 ± 4.38 | 32.64 ± 4.86 | 0.211 |
| CEC | 11.64 ± 1.56 | 16.8 ± 6.04 | 0.051 | 14.29 ± 5.78 | 15.33 ± 8.45 | 0.691 |
| Base saturation | 0.47 ± 0.1 | 0.64 ± 0.13 | 0.015* | 0.01 ± 0 | 0.05 ± 0.06 | 0.006** |
| pH | 3.93 ± 0.13 | 3.86 ± 0.27 | 0.496 | 5.24 ± 0.21 | 5.02 ± 0.31 | 0.014* |
| $Fe_{SRO}$ | | | | 5.03 ± 2.35 | 2.07 ± 1.5 | 0.003** |
| $Al_{SRO}$ | | | | 8.75 ± 4.6 | 1.39 ± 2.72 | 0.000*** |
| $Fe_{CRI}$ | | | | 1.55 ± 1.27 | 0.86 ± 1.30 | 0.089 |





**Table 3: Intragroup community indexes of soil physico-geochemical variables relatively to orthogonal canonical components of vegetation variables**

|  | t1 | t2 | t3 |
|---|---|---|---|
| $Al_{oxa}$ | **0.38** | **0.43** | **0.73** |
| $Fe_{oxa}$ | 0.17 | 0.24 | 0.35 |
| $Fe_{dit}$ | 0.16 | 0.24 | 0.41 |
| $Al_{dit}$ | 0.18 | 0.29 | **0.70** |
| $Fe_{SRO}$ | **0.37** | **0.37** | **0.71** |
| $Al_{SRO}$ | **0.38** | **0.38** | **0.56** |
| Ca | **0.28** | **0.35** | 0.46 |
| Mg | 0.23 | 0.28 | **0.97** |
| Fe | **0.51** | **0.52** | 0.28 |
| B horizon thickness | 0.13 | 0.14 | 0.22 |
| pH | 0.06 | 0.13 | 0.35 |

**Table 4: Intragroup community indexes of vegetation variables relative to orthogonal canonical components of soil physico-geochemical variables**

|  | u1 | u2 | u3 |
|---|---|---|---|
| Basal area | 0.55 | 0.55 | 0.59 |
| Moss cover | 0.43 | 0.49 | 0.53 |
| Lichen cover | 0.43 | 0.49 | 0.53 |

10 **Table 5: Results of permANOVA on the ecosystem types and geochemical variables (MF *vs* LW).** Significant effects are indicated in bold. P(MC): p-values obtained through Monte Carlo methods.

| Source | df | MS | Pseudo-F | P(perm) | P(MC) | Number of perm. units | % of variance explained |
|---|---|---|---|---|---|---|---|
| Ecosystem type (MF vs LW) = (ty) | 1 | 92.103 | 12.939 | **0.0001** | **0.0005** | 10 | 47.6 |
| Site (ty) = si(ty) | 4 | 7.1183 | 1.3902 | 0.2339 | 0.2115 | 9928 | 3.4 |
| Transect (tr) = tr(si(ty)) | 12 | 5.1203 | 2.2214 | **0.0006** | **0.0017** | 9887 | 14.2 |
| Residuals | 36 | 2.305 |  |  |  |  | 34.8 |
| Total | 53 |  |  |  |  |  | 100 |





**Table 6: Carbon and nitrogen stocks contained in FH and B horizons.** Values are given as means ± standard deviations.

| | | Total stock at the site scale (t.ha$^{-1}$) | |
| --- | --- | --- | --- |
| | | LW sites | MF sites |
| FH horizon | | | |
| | C | 36.68 ± 4.29 | 126.44 ± 67.86 |
| | N | 0.53 ± 0.05 | 2.06 ± 1.00 |
| B horizon | | | |
| | C | 22.23 ± 8.55 | 44.68 ± 6.61 |
| | N | 0.67 ± 0.31 | 1.39 ± 0.32 |