# Peer review of "Boreal coniferous forest density lead to significant variations in soil physical and geochemical properties"

_Biogeosciences, 2017_

## Referee Comment (RC1) · Anonymous Referee #1 · 31 Mar 2017

General comments

This paper examines the differences in soil properties between two forest ecosystems in north eastern Canada and draws conclusions as to the likely drivers of these differences. The authors conclude that the observed differences in soil properties can be largely attributed to the impact of different vegetation cover, which in turn is related to changes in the frequency of forest fires over the past millennia. The paper is very clearly written, the results are clearly presented and the discussion is well argued and supported by literature.

Specific comments

1. Does the paper address relevant scientific questions within the scope of BG?

The paper clearly addresses scientific questions that are within the scope of the Biogeosciences journal and will be of interest to the journal readers.

2. Does the paper present novel concepts, ideas, tools, or data?

The paper is well founded on the existing understanding of soil paedogenesis and the factors driving the variability of soil physical and chemical properties. The paper concludes with a clear and enlightening synthesis of the detailed findings, which are presented in a clear conceptual diagram. This new conceptual framework is an important contribution as it opens the way for further hypothesis testing in subsequent research.

However, on initial reading, the novelty of the paper is not clear from the outset. On page 5, lines 21-23 it is stated that 'This experimental design was conceived for further investigations linking soil and lake sediment composition at the watershed scale. Indeed, it corresponds to the first step of a paleoecological investigation aiming to retrace the opening of the landscape over time using geochemistry analysis from lacustrine deposits.' If this paper is a first in a series and provides a methodological underpinning to subsequent work, I suggest that this should be brought to the readers' attention at the outset so that the purpose of the work is clear from the beginning.

3. Are substantial conclusions reached?

Substantial conclusions are reached in respect of the influence of vegetation on the development of soil properties and the consequences of these changes for forest management and soil carbon sequestration.

4. Are the scientific methods and assumptions valid and clearly outlined?

The scientific methods are clearly outlined. However, I would propose to include additional Figures in the paper that would i) clearly show the location of the study site within Canada/North America ii) show a schematic of the experimental sampling design and

particularly the different methods of sampling individual soil horizons (mentioned in the Methods section page 6, lines 0-11)

5. Are the results sufficient to support the interpretations and conclusions?

Yes

6. Is the description of experiments and calculations sufficiently complete and precise to allow their reproduction by fellow scientists (traceability of results)?

Methods could be further clarified on page 6, lines 0-11. How were soil samples collected from the mineral B and C horizons? What was the diameter of the soil auger used? How was bulk density calculated – presumably this was only done for the volumetric samples?

7. Do the authors give proper credit to related work and clearly indicate their own new/original contribution?

Yes

8. Does the title clearly reflect the contents of the paper?

The title could be made more interesting/dynamic, reflecting the purpose and the research in respect of the most interesting conclusions that only become clear at the end of the manuscript. These include significant conclusions pertaining to the vegetation/soil interactions, forest management and the implications for soil carbon stocks and global change.

9. Does the abstract provide a concise and complete summary?

The abstract does not fully make clear the novelty of this work. In fact, the novelty only becomes apparent after a thorough reading of the full manuscript. I suggest that the authors bring forward their most important conclusions and include these in the abstract and ideally also more clearly prime the reader to the purpose of the work at the end of the Introduction.

[Figure]

10. Is the overall presentation well structured and clear?

Yes

11. Is the language fluent and precise?

Yes

12. Are mathematical formulae, symbols, abbreviations, and units correctly defined and used?

Yes

13. Should any parts of the paper (text, formulae, figures, tables) be clarified, reduced, combined, or eliminated?

No

14. Are the number and quality of references appropriate? Yes

15. Is the amount and quality of supplementary material appropriate?

Yes

Technical corrections

Page 4, line 25 – can you please specify the size of the study sites?

Page 6, lines 3-5 sentence 'They were composed..' should perhaps be part of the results, rather than methodology?

Page 12, line 2 – should read 'did not find'

Page 12, line 3 – can you please give some examples of references after the statement 'in line with those of other studies'

Page 13, line 11 – should read 'environmental conditions'

---

## Referee Comment (RC2) · Anonymous Referee #2 · 24 Apr 2017

General comments: The authors investigated the impact of an ecosystem shift in boreal forests from closed- canopy moss forests to open lichen woodlands on soil carbon and nutrients contents as well as Fe and Al crystallinity in soil profiles down to 80 cm deep. The study was well planned and properly conducted. The data is clearly presented and the discussion is very well structured. The writing is very fluent and logical. The reader never has a feeling of being lost or confused despite the large amount of parameters determined. All in all the manuscript was a very enjoyable read and I recommend the mansucript for publication without changes except for a few typos and omissions listed below.

(Personal note: I prefer consecutive line numbering because it is less cumbersome to

[Figure]

refer to specific lines in a review as one does not need to include the page number as well.)

Specific comments: None

Technical comments: Abstract line 13: "...that determines the of vegetation stucture..." – there's a word missing (probably "type of vegetation structure") Materials and Methods pg 5, line 1: I stumbled over this sentence at first as "annual sum" made me automatically assume "annual mean", which of course didn't work with the 1186.4 days. I assume this is the sum of GDD over the whole 1981-2010 period? Maybe the sentence could benefit from a slight rephrasing/clarification. pg 5, line 19: If I understood your site description correctly there should be three EU per given forest type not six. pg 7, line 9: I suggest to add "(SRO)" behind "short-range order" in order to make the abbreviation explanation easier to find for readers who don't read the manuscript in the "correct" order (i.e. looking at figures and results first etc.)

Results & Discussion pg 12, line 2: "find" instead of "found"

Figures Figures 2 to 4: Complete the figure captions for each figure and do not refer to the legend of figure 1. Depending on type setting of the manuscript later on the figures might end up on different pages, which would then force readers to flip pages back and forth.

Tables Table 1: "Eleva-tion" instead of "Eleva-Tion" move "(%)" behind soil texture to sand, silt and clay in order to be consistent with the preceeding columns (% moss . . ..) Table 2: add units (horizon thickness, CEC, etc.) Table 3: define the meaning of the bold values Table S1: add units

1. Does the paper address relevant scientific questions within the scope of BG? Y 2. Does the paper present novel concepts, ideas, tools, or data? Y 3. Are substantial conclusions reached? Y 4. Are the scientific methods and assumptions valid and clearly outlined? Y 5. Are the results sufficient to support the interpretations and conclusions?

Y 6. Is the description of experiments and calculations sufficiently complete and precise to allow their reproduction by fellow scientists (traceability of results)? Y 7. Do the authors give proper credit to related work and clearly indicate their own new/original contribution? Y 8. Does the title clearly reflect the contents of the paper? Y 9. Does the abstract provide a concise and complete summary? Y 10. Is the overall presentation well structured and clear? Y 11. Is the language fluent and precise? Y 12. Are mathematical formulae, symbols, abbreviations, and units correctly defined and used? Y 13. Should any parts of the paper (text, formulae, figures, tables) be clarified, reduced, combined, or eliminated? N 14. Are the number and quality of references appropriate? Y 15. Is the amount and quality of supplementary material appropriate? Y

---

## Author Comment (AC1) · 24 Apr 2017

Thank you very much for all your constructive comments and suggestions. They will all be taken into consideration in the revised version.

In particular, we will make sure that the revised manuscript provides more details on the study area (we will extend the location on Figure S1 and move it from supplementary material to the main paper), and we will be add details on the sampling soil sampling methodology.

We will emphasize the importance of our work and mention the context of this work sooner in the text to give a broader picture of our research. We will highlight the

novelty of our findings and modify the abstract consequently. Thank you very much for reviewing our paper.
* * *

---

## Author Comment (AC2) · 24 Apr 2017

Thank you very much for your considerate comments and for the edition suggestions. We will change the figure captions accordingly and incorporate all the suggestions from the other reviewer in the final version. Thank you for reviewing our manuscript.
* * *

---

## Referee Comment (RC3) · Anonymous Referee #3 · 20 May 2017

General Comments

This manuscript investigated different soil properties and soil horizon development between boreal closed-canopy forests (MF) and open lichen woodlands (LW). The authors found more organic carbon and exchangeable base cations (Ca, Mg) in MF humus than LW soil. The B horizon of LW sites contained more amorphous Fe/Al oxides than MF mineral soils. The authors have done solid research work about detailed comparison between two forest ecosystems for soil chemical properties, Fe/Al reactive species, covariance between vegetation and soil geochemical variables, and so on. The manuscript is easy to follow, well written and logically structured. However, some weaknesses and discussion need to be improved. The title of this paper is "soil

carbon, available nutrients, and iron and aluminium crystallinity vary between boreal closed-canopy forests and open lichen woodlands". However, there is rare discussion about why available nutrients vary between these two forests. Although section 3.4 discussed P distribution, it only focused on the comparison among different soil horizons not between the two forest soils. Also, the authors emphasized the difference of iron/aluminum between two forest systems and attributed it to different pedogenetic development under MF and LW cover. However, there is no deeper discussion about how the pedogenetic development accumulated more amorphous Fe/Al in B horizon of lw than mf. Additionally, the novelty and uniqueness of this paper are not clear in the introduction.

Other special comments:

Comment 1: the title could be improved. As mentioned before, the comparison of available nutrients are rarely discussed.

Comment 2: Page 2 line 9: The authors mentioned lw soils were nutritionally poorer. Which kind of nutrient do you mean here? If authors emphasized phosphorus, the concentration of phosphorus in FH horizons does not have significant difference between two forest ecosystems.

Comment 3: Page 4, line 19-21. The assumption is no clue from former description.

Comment 4: Page 5. Line 1: It is confused that annual sum was 1186.4 and please make it clear.

Comment 5: section 2.3.1. Could you provide more details about the C and N contents measurement?

Comment 6: Section 2.3.2. Please give explanation about which fraction of ions and phosphorus could be extracted by the Mehlich-3 solution.

Comment 7: Page 7 Line 5-6: "Quantities of Al extracted by oxalate (Aloxa) may be higher than quantities extracted by dithionite citrate (Aldit) in some cases such as in

acid soils or podzols." Please explain the reason.

Comment 8: Section 3.1: Provide further explanation how the different concentration of Fe oxides and vegetation cover between lw and mf plots lead to the light yellow color in lw plots and darker and browner color in mf plots.

Comment 9: Section 3.2: Please give more discussion why extractable phosphorus is higher in lw C horizon than in mf C horizon, and why there is no significant difference of phosphorus in B and FH horizons between these two forest soils. The authors only mentioned that the accumulation of products of mineral weathering as well as the migration of organic P compounds could explain this different, but it is still not clear for audiences.

Comment 10: in section 3.4, the authors discussed the P distribution among different soil horizons and found lower concentration of extractable P in B horizon than FH and C horizons. Have you done the statistical analysis about it? Is there significant difference between B horizon and FH/C horizon? Also there is no explanation why FH horizon have more P than B horizon. What is the relation between the enrichment in organic C and high concentration of extracted P in soil?

Comment 11: Page 11, line 7-10: "The different behaviours of exchangeable Fe and bound Fe could be explained by their different mobility properties and abilities, in particular since fluxes could vary under different soil environment conditions and thicknesses between lw and mf plots." I think authors should give further discussion how different fluxes and thicknesses between lw and mf plots affect the mobility properties and abilities of exchangeable Fe and bound Fe.

Technical comments:

Page 8 line 18: "(Fig. 1b) and 1c)". Delete the ")" in the middle.

Page 12 line 1: "found" changed to "find".

Fig. 4: give explanation about Alpyro, Aloxa, Aldit, AlSRO, Fepyro, Feoxa, Fedit,

FeSRO behind the figure or on the figure caption.

Table 2: provide the explanation about CEC, FeSRO, AlSRO and FeCRI on the table caption.

Table S1: provide the units of P and metal species.

———————————————

---

## Author Comment (AC3) · 2 Jun 2017

Dear referee,

Thank you very much for your constructive comments and suggestions for improvement and clarification.

As suggested, in the revised manuscript we plan to emphasize the novelty of our approach and give more precise details about the context of our work at the end of the introduction.

Here are individual responses to your comments and some details about the revisions

we plan to make for a final acceptance:

- Comment 1: We are working on a title change in order to make it more impactful and we won't mention available nutrients.

- Comment 2: The term nutritionally poorer generates confusion and will be removed.

- Comment 3: We will make the sentence clearer and add elements for clarifying our assumption Revised version: "we also hypothesized that the podzolisation process and hence iron reactive chemical species would be different depending on the local vegetation density. We assumed that the conditions found in soils covered by a lichen mat with low-density canopy would be more prone to iron and aluminium oxide accumulations than MF soils. Schaetzl et al. (2015) argued that water fluxes have a great influence on the intensity of podzolization especially because they control the mobility of soluble organic complexes onto the soil profiles. Snow, snowmelt and deep percolation may thus vary between MF and LW because of tree density and lichen/moss cover.

- Comment 4: We corrected the info about the number of degree days to make it clearer Revised version: "The number of degree days above $5°C$ for the period 1981-2010 was 817 per year"

- Comment 5: We will add details about the methodology used for C and N contents measurement Revised version: "Total C and N contents (%) were measured on all soil samples by combustion using an induction furnace (Leco® TruMac CNS Analyzer) following sieving at 2mm, drying and ground at 0.5 mm. The combustion is performed at $1350°C$ under an oxygen gas atmosphere which turns C and N forms to CO2, N2 and NOx. Gaz concentrations are the determined by thermal conductivity and infrared detection."

- Comment 6: We will add some info. Mehlich is a well know method that is mainly used to determine exchangeable cations. The fluoride contained in ammonium fluoride promotes P desorption by decreasing Al activity and by forming Al-Fl complexes. It is

appropriate for assessing P availability in acidic soils. Revised version: "The Melich-3 solution mainly extracts exchangeable and soluble cations and phosphorus under aluminium, calcium and iron phosphate forms."

- Comment 7: We are not entirely aware of the reasons: as noted by the authors (Johnson and Todd, 1983; Pagé and Kimpe, 1989) Aloxa is high in these soils due to the importance of amorphous forms of Al as was as that of Al-organic matter complexes. Revised version: "McKeague et al. (1971) showed that these two extractants are less useful in distinguishing species of Al in soils than for Fe species."

- Comment 8: Soils in MF contain more organic matter, which explain the darker horizons. We will add some info about colorations and Fe oxides concentrations.

Revised version: "The more important the pedogenic process of Fe reduction and subsequent removal are, the less colourful the soils are, displaying mostly the grey colours of the silicate matrix (Schwertmann, 1985). It is thus likely that pedogenic processes differ between lw and mf plots."

- Comments 9 and 10: P was not the focus of this study and in fact it did not strike out as being linked to the type of forest studied. We measured extractable P in the C horizon to evaluate if the parent material was different between the two environments. The difference that appeared here was minor and, as it was higher in LW suggested that the differences were minor enough that it did not play a role for ecosystem development.

- Comment 12: As suggested, we will improve our discussion and give further elements to discuss why Fe and Al oxides could behave differently depending on the pedoenvironmental conditions. Revised version: "The conversion reactions of iron oxides depend to a large extent to pedoenvironmental factors (pH, water activity, temperature, etc.) (Schwtermann, 1988). These factors vary with depth and depend on the groundcover. Furthermore, organic matter seems to have an influence on iron oxides by inhibiting their crystallinity (Borggaard et al., 1990): in mf plots, the thicker and denser organic matter layer could explain the lower concentrations of Fe and Al oxides in B

horizons. Fe and Al oxides species could also differ between mf and lw plots because soil temperature and moisture are also responsible for different goethite:hematite ratios (Schwtermann, 1988). Hematic soils develop in warmer conditions and are characterized by reddish brown colors while goethitic soils develop under colder environment and turn yellowish-brown. This is consistent with our observations of clearer red to yellow soils under lw cover where little organic matter accumulates as opposed to mf soils overlay by a thick dark brown organic layer which could lead to warmer temperatures."

---

## Author Response (AR1)

Dear Editors,

Following your decision on providing our manuscript with minor revisions, we would like to give some details about the changes that we made according to the reviewers' comments and suggestions.

**Point-by-point response to the reviews**

- Response to Reviewer#1's general comments:

We highlighted the novelty of the paper sooner in the introduction and emphasized the purpose of our work constituting a first step approach for further paleological investigations (p4 of the revised version – non marked-up-, l.14-17). We moved the figure showing our study area to the main manuscript rather than supplementary material and added precisions on their location within North America. We completed our methods description for soil sampling process with a new figure, added to the supplementary material (Fig. S1). We worked on a more relevant title and brought the main conclusions of our work forward to the abstract to highlight the novelty of our work.

- Response to Reviewer#2's general comments:

We made all the revisions suggested in the reviewer's technical comments

- Response to Reviewer#3's general comments:

Consistently to what we said in the response made to reviewer#1, we emphasized the novelty of our approach and gave more specific details about the context of our work at the end of the introduction. Concerning the specific comments, here are some individual responses (sentences in italic were added to the revised version of the manuscript):

Comment 1: We worked on a title change in order to make it more impactful. It does not mention available nutrients anymore.

Comment 2: The term nutritionally poorer generated confusion and was removed (Abstract, l.12).

Comment 3: We made the sentence clearer and added elements for clarifying our assumption (p4, l.21-26).

*"we also hypothesized that the podzolisation process and hence iron reactive chemical species would be different depending on the local vegetation density. We assumed that the conditions found in soils covered by a lichen mat with low-density canopy would be more prone to iron and aluminium oxide accumulations than MF soils. Schaetzl et al. (2015) argued that water fluxes have a great influence on the intensity of podzolization especially because they control the mobility of soluble organic complexes onto the soil profiles. Snow, snowmelt and deep percolation may thus vary between MF and LW because of tree density and lichen/moss cover."*

Comment 4: We corrected the info about the number of degree days to make it clearer (p5, l.7).

*"The number of degree days above 5∘C for the period 1981-2010 was 817 per year"*

Comment 5: We added details about the methodology used for C and N contents measurement (p6, l.19-22).

*"Total C and N contents (%) were measured on all soil samples by combustion using an induction furnace (Leco® TruMac CNS Analyzer) following sieving at 2mm, drying and grinding at 0.5 mm. The combustion is performed at 1350∘C under an*

*oxygen gas atmosphere which turns C and N forms to CO2, N2 and NOx. Gaz concentrations are then determined by thermal conductivity and infrared detection."*

Comment 6: We added some info on Mehlich extractions (p6, l.28-29).

*"The Melich- 3 solution mainly extracts exchangeable and soluble cations and phosphorus under aluminium, calcium and iron phosphate forms."*

Comment 7: We specified that McKeague et al. (1971) observed differences in Al and Fe extractions by oxalate and dithionite (p7, l.17-18).

*"McKeague et al. (1971) showed that these two extractants are less useful in distinguishing species of Al in soils than for Fe species."*

Comment 8: We added some info about colorations and Fe oxides concentrations (p8, l.32-p9, l.2).

*"The more important the pedogenic process of Fe reduction and subsequent removal are, the less colourful the soils are, displaying mostly the grey colours of the silicate matrix (Schwertmann, 1985). It is thus likely that pedogenic processes differ between lw and mf plots."*

Comments 9 and 10: P was not the focus of our study and in fact it did not strike out as being linked to the type of forest studied. We measured extractable P in the C horizon to evaluate if the parent material was different between the two environments. The difference that appeared here was minor and, as it was higher in LW suggested that the differences were minor enough that it did not play a role for ecosystem development. We chose not to add new material on this matter.

Comment 11: we improved our discussion and gave further elements to discuss why Fe and Al oxides could behave differently depending on the pedoenvironmental conditions (p11, l.23-32).

*"The conversion reactions of iron oxides depend to a large extent to pedoenvironmental factors (pH, water activity, temperature, etc.) (Schwtermann, 1988). These factors vary with depth and depend on the groundcover. Furthermore, organic matter seems to have an influence on iron oxides by inhibiting their crystallinity (Borggaard et al., 1990): in mf plots, the thicker and denser organic matter layer could explain the lower concentrations of Fe and Al oxides in B C3 BGD Interactive comment Printer-friendly version Discussion paper horizons. Fe and Al oxides species could also differ between mf and lw plots because soil temperature and moisture are also responsible for different goethite:hematite ratios (Schwtermann, 1988). Hematic soils develop in warmer conditions and are characterized by reddish brown colors while goethitic soils develop under colder environment and turn yellowish-brown. This is consistent with our observations of clearer red to yellow soils under lw cover where little organic matter accumulates as opposed to mf soils overlay by a thick dark brown organic layer which could lead to warmer temperatures."*

**Relevant changes made in the manuscript**

- The title was changed to: "Boreal coniferous forest density lead to significant variations in soil physical and geochemical properties"
- The novelty of our work was brought forward in the introduction and abstract, as well as the purpose of our work
- Figure S1 was moved to the main manuscript rather than being placed in supplementary material and made more detailed
- A figure on soil sampling methods was added to the supplementary material
- Some pieces of information were added, mainly in the discussion, to respond to the third reviewer's need of clarifications concerning biogeochemical processes.
- A link to the data file was added the part "5 Data availability"

**Marked-up manuscript version:**

starting form next page

[revised manuscript text omitted]

**Commented [cb2]:** Reply to R2 : We moved this part to the description of Fig.2 in the results.

**Commented [cb3]:** Reply to R1: We removed this info because we did not use bulk density in our analyses

[revised manuscript text omitted]